# Prognostic Factors for the Efficiency of Radiation Therapy in Dogs with Oral Melanoma: A Pilot Study of Hypoxia in Intraosseous Lesions

**DOI:** 10.3390/vetsci10010004

**Published:** 2022-12-22

**Authors:** Shunsuke Noguchi, Kohei Yagi, Nanako Okamoto, Yusuke Wada, Toshiyuki Tanaka

**Affiliations:** 1Laboratory of Veterinary Radiology, Graduate School of Veterinary Science, Osaka Metropolitan University, 1-58 Rinku Ourai Kita, Izumisano-shi 598-8531, Osaka, Japan; 2Laboratory of Veterinary Radiology, College of Life, Environment, and Advanced Sciences, Osaka Metropolitan University, 1-58 Rinku Ourai Kita, Izumisano-shi 598-8531, Osaka, Japan; 3Veterinary Medical Center, Osaka Metropolitan University, 1-58 Rinku Ourai Kita, Izumisano-shi 598-8531, Osaka, Japan

**Keywords:** bone lysis, hypoxia-inducible factor-1α, local recurrence, oral melanoma, radiation therapy

## Abstract

**Simple Summary:**

Prognostic factors other than the clinical stage for canine oral melanoma treated with radiation therapy remain unclear. In our cohort, debulking surgery prior to radiation therapy was significantly associated with good response to radiation therapy. Furthermore, the presence of bone lysis was associated with worse response to radiation therapy, which might be due to hypoxia in the osseous tissue.

**Abstract:**

Unresectable oral melanoma is often treated with radiation therapy (RT) and may show a temporary response to therapy. The clinical stage is one of the well-known prognostic factors for canine oral melanoma. However, the factors that directly affect the response to RT have remained unclear. This study aimed to validate the risk factors for recurrence after RT. Sixty-eight dogs with oral melanomas were included in this study. All dogs were treated with palliative RT using a linear accelerator without adjuvant therapies. After RT, the time to local recurrence (TTR) and overall survival (OS) were evaluated using the log-rank test. As a result, clinical stage and response to therapy were the significant independent prognostic factors in the multivariate analysis. The presence of local bone lysis and non-combination with cytoreductive surgery were associated with a worse response to RT. Immunohistochemical analysis for hypoxia-inducible factor-1α indicated that tumor cells invading the bone are under hypoxic conditions, which may explain a poorer efficiency of RT in dogs with bone lysis. In conclusion, clinical stage and combination with debulking surgery were needed to improve the efficiency of RT.

## 1. Introduction

Oral melanoma is one of the most common malignancies in dogs [1,2]. Unresectable oral melanomas are often treated with radiation therapy, although the first-line treatment is surgical resection [3]. A recent study suggested that the efficiency of radiation therapy is almost equivalent to that of surgical treatment for advanced canine oral melanoma [3]. However, recurrence after radiation therapy is common. Several prognostic factors, such as tumor size, bone destruction, and clinical stage, to predict overall survival time have been identified in dogs with oral melanoma who have undergone radiation therapy [3,4,5]. Of those, the influence of tumor location and bone destruction on the efficiency of radiation therapy is a topic of debate, because there are few reports regarding time to recurrence after radiation therapy. In addition, the molecular mechanisms that influence sensitivity to irradiation have also been experimentally validated in recent studies [6]. However, the clinical factors predicting the efficiency of radiation therapy have not yet been defined.

The cytotoxic effects of irradiation can be divided into direct and indirect effects. The indirect effects of irradiation, which mainly contribute to cytotoxic reactions, require oxygen to produce free radicals [7]. Thus, tumor cells in the hypoxic region are considered relatively resistant to irradiation [8,9,10,11]. Tumor cells commonly receive oxygen from the blood through vessels. However, recent studies have shown that bone parenchyma and bone marrow are under relatively hypoxic conditions in comparison with periosteum [12,13]. Thus, tumor cells invading into the bone can potentially acquire resistance to irradiation as a result of hypoxia.

Hypoxia is commonly present in the microenvironment of solid tumors including melanoma [14,15]. Hypoxia-inducible factor-1α (HIF-1α) is one of the most important factors mediating the adaptive response to hypoxia in tumor tissue [16,17]. HIF-1α is stabilized under hypoxic conditions, although it is fragile under normoxic conditions. Stabilized HIF-1α translocates to the nucleus and induces the expression of several genes [18]. In human cancer, it has been indicated that HIF-1α is associated with resistance to radiation therapy [19,20]. In dogs, the expression of HIF-1α is increased in several kinds of cancer and has been considered to be a therapeutic target, as well as human cancers [21,22,23].

This retrospective study explored whether factors such as bone lysis, tumor location, tumor size, and clinical stage affect the response to irradiation in canine oral melanoma. Furthermore, to validate the independent factors affecting the response to irradiation, multivariate analysis was performed. In addition, HIF-1α expression was evaluated in surgically resected oral melanoma tissues. This study may contribute to the appropriate determination of therapies for canine oral melanoma.

## 2. Materials and Methods

### 2.1. Case Selection

The medical records of the veterinary medical center at Osaka Prefecture University were searched to identify dogs diagnosed with oral melanoma and treated with radiation therapy between 1 April 2017 and 31 December 2021. Dogs that underwent radiation therapy with or without cytoreductive surgery were included, and those that received adjuvant chemotherapy and those that underwent incomplete radiation therapy were excluded. Dogs with unknown survival times were also excluded.

### 2.2. Medical Record Review

The recorded information included signalment, results of diagnostic imaging examinations and biopsies, clinical stage at the time of first irradiation, and follow-up information, including time to local recurrence (TTR; the time between the first radiation treatment and the date of recurrence) and overall survival (OS; the time between the first radiation treatment and the date of death). The dates of recurrence and death were clarified by medical records or telephone calls with the owners or referral doctors. Utilization of medical records and sample correction were approved by the Ethical Review Board of the Animal Medical Center of Osaka Prefecture University (Approval number: R01-001) and the dog owners. All applicable international, national, and/or institutional guidelines for the care and use of animals were followed.

### 2.3. Evaluation of Clinicopathological Status

General anesthesia was induced using propofol and isoflurane. All dogs underwent computed tomography (CT) to determine the clinical stage and plan treatment before the first radiation treatment. The entire body was scanned using CT. Biopsy specimens were collected by excisional biopsy. Any enlarged local lymph nodes (mandibular lymph nodes and retropharyngeal lymph nodes) were also biopsied using a fine needle. However, lymph nodes that were subjectively assessed as normal in size were not evaluated. All the biopsy samples were histopathologically evaluated by board-certified veterinary pathologists. Tumors were classified as stages I, II, III, or IV according to the World Health Organization staging guidelines [1]. The presence of local bone lysis, which was evaluated based on the CT images (Appendix A), was not considered in the classification of clinical staging. Tumors were classified according to their anatomical location (caudal mandibular gingiva, caudal maxillary gingiva, rostral region, lip, and hard or soft palate). The rostral region was defined as the region rostral to the fourth premolar of the mandible and maxilla.

### 2.4. Radiation Therapy

All dogs underwent radiation therapy. Treatment was delivered using a linear accelerator with an X-ray energy output of 4 MV (Primus Mid Energy; Canon Medical Systems, Tochigi, Japan). For radiation therapy, planning CT scans were performed for each dog in the treatment position with a bite block and vacuum-mattress immobilization device. Treatment plans were constructed using a 3D CT-based computer-generated treatment planning system (Xio; Elekta Japan, Tokyo, Japan). The gross tumor volume (GTV) was defined by the contrast-enhancing area on CT images. The clinical target volume (CTV) was contoured from 0.3 to 0.8 cm of GTV to include regions at risk for microscopic disease. Then, the CTV margin was extended three dimensionally by 0.2 cm to define the planning target volume (PTV), accounting for safety margins for positioning errors. In dogs that underwent cytoreductive surgery before radiation therapy, the CTV was contoured based on the location and extent of the tumor. Subsequently, the PTV margins were contoured from 0.2 cm around the CTV limits. The isocenter and beam arrangements for each plan were determined on the basis of the location of the tumor and adjacent critical normal structures. Treatments were delivered in parallel in opposing fields or in four fields in multiple beam arrangements. The energy output of the X-ray beam was 4 MV. The dose prescription was 8.0 Gy in four fractions, for a total dose of 32 Gy. Irradiation was performed at 7-day intervals. Portal imaging was performed before the first treatment session to ensure appropriate patient positioning.

### 2.5. Evaluation of Treatment Response

The response to treatment was determined at the time of achieving maximum reduction in tumor size, which was assessed by CT examination according to the classification of the Veterinary Cooperative Oncology Group RECIST guidelines for dogs [24]. Appearance of metastatic lesions to lymph nodes and distant organs was excluded from the PD criteria because this study focused on the efficiency of local control by radiation therapy.

### 2.6. Immunohistochemistry

Immunohistochemistry (IHC) was performed on five formalin-fixed paraffin-embedded canine oral melanoma tissues with bone lysis to evaluate the expression of HIF-1α. All the tissues were surgically resected and decalcified after formalin fixation. The antigen-retrieval procedure included autoclave pretreatment at 121 °C for 15 min in 0.01 M citrate buffer (pH 6.0). After inhibition of nonspecific reactions with 5% skim milk and 0.1% NaN_3_ in phosphate-buffered saline for 15 min at 25 °C (room temperature), the slides were incubated overnight at 4 °C with the rabbit polyclonal primary antibody against HIF-1α (Bethyl Laboratories, Montgomery, TX, USA) diluted 200-fold, which was referred to in the previous study [25]. The secondary antibody consisted of rat anti-rabbit IgG conjugated to biotin (Histofine Simple Stain Rat MAX-PO [R]; Nichirei, Tokyo, Japan). Negative controls were prepared using an irrelevant isotype-matched antibody (Rabbit IgG, polyclonal-Isotype control; Abcam, Cambridge, UK) instead of the primary antibodies. Immunostaining was visualized using 3,3′-diaminobenzidine tetrahydrochloride (Peroxidase Stain DAB Kit; Nacalai Tesque). The slides were then washed with distilled water and counterstained with Giemsa solution to distinguish the positive staining from melanin granules. To assess HIF-1α immunoreactivity, positive staining of the nucleus was defined as positive and the intraosseous positive rate (the number of nucleus-positive tumor cells/all of the number of tumor cells) was calculated, including the marrow cavity and extraosseous region, respectively. The positive rate in five high-power fields was then averaged.

### 2.7. Statistics

Kaplan–Meier survival curves were generated for all dogs; those surviving at the end of the investigation period or deceased by causes unrelated to melanoma progression were censored. Similarly, dogs that experienced no recurrence at the date of death were censored for the TTR analysis. Significant differences in survival time were analyzed by log-rank tests using GraphPad Prism software (USACO, Tokyo, Japan). Differences in body weight, age, sex, and tumor location were evaluated by the χ^2^ test, and those in response to treatment were assessed by the Mann–Whitney test using Excel software (Microsoft, Redmond, WA, USA). The association between treatment and TTR was evaluated with the multivariate Cox proportional hazards model using the Easy R software (Jichi Medical University, Tochigi, Japan) for significant variables (*p* < 0.05) identified in the univariate analysis. The difference in the positive rate of HIF-1α expression was evaluated using a two-tailed paired *t*-test. Statistical significance was set at *p* < 0.05.

## 3. Results

### 3.1. Patient Characteristics

Sixty-eight dogs from eighteen breeds were included in this study. Miniature dachshunds (*n* = 26 (38.2%)) and toy poodles (*n* = 9 (13.2%)) were the most commonly affected breeds, while the other breeds were as follows: Shiba (*n* = 6); Mongolian (*n* = 6); Labrador Retriever (*n* = 3); Papillon (*n* = 2); Pekingese (*n* = 2); Beagle (*n* = 2); Pomeranian (*n* = 2); Yorkshire Terrier (*n* = 1); American Cocker Spaniel (*n* = 1); Pumi (*n* = 1); Chihuahua (*n* = 1); Basenji (*n* = 1); Shih Tzu (*n* = 1); Welsh Corgi (*n* = 1); Miniature Schnauzer (*n* = 1); St. Bernard (*n* = 1); and Cavalier King Charles Spaniel (*n* = 1). The median body weight of the dogs was 5.8 kg (range: 1.7 to 65 kg), and the median age at the first radiation therapy was 13 years (range: 4 to 18 years). The study population included 11 intact males, 23 castrated males, seven intact females, and 27 spayed females. No significant difference was observed between the sexes (*p* = 0.28). All of the patient data are indicated in Appendix A.

### 3.2. Tumor Characteristics in Relation to Clinical Stage

Stage I, II, III, and IV disease was observed in seven (10.3%), 14 (20.6%), 29 (42.6%), and 18 (26.5%) dogs, respectively. No significant differences were observed in the weight, age, or sex of the dogs according to the clinical stage (Table 1). In contrast, the study population included 39 dogs with bone lysis and 29 without bone lysis, and the number of dogs with bone lysis significantly increased as the clinical stage advanced (*p* = 0.00052). Twenty-one dogs underwent debulking surgery, and the number of dogs with cytoreduction significantly decreased as the clinical stage advanced (*p* = 0.000030). The locations of melanomas included the maxilla (*n* = 23), mandible (*n* = 30), rostral region (*n* = 5), buccal region, lip, and hard or soft palate (*n* = 10).

### 3.3. Outcomes

The median TTR and OS for all 68 dogs were 133 and 144.5 days, respectively. There were not any significant differences between TTR and OS (*p* = 0.45). The median TTR and OS for dogs in each clinical stage were as follows: TTR, stage I, 342 days; stage II, 157 days; stage III, 126 days; and stage IV, 91 days; OS, stage I, not reached; stage II, 234; stage III, 142 days; and stage IV, 93 days. There were also not any significant differences between TTR and OS of each clinical stage. Survival curves are shown in Figure 1. The log-rank test revealed a significant difference between each group except for stages II and III in TTR and stages I and II in OS. Seven dogs were still alive at the end of the study period (three with stage I disease (1102, 375, and 202 days) two with stage II disease (910 and 144 days), and two with stage III disease (197 and 128 days)).

No significant differences were observed in TTR and OS (data not shown) in relation to tumor location. The median TTR and OS for dogs in relation to tumor size were as follows: TTR, TI, 158 days; T2, 148 days; and T3, 126 days; and OS, T1, 312 days; T2, 234 days; and T3, 145 days (Figure 2a). The log-rank test revealed a significant difference between T1 and T3 in the TTR and each group, except for T1 and T2, in the OS. Cytoreduction by debulking surgery significantly affected the TTR and OS: the TTR was 179.5 days with cytoreduction and 119 days without cytoreduction, while the OS was 312 days with cytoreduction and 148 days without cytoreduction (Figure 2b).

The presence of bone lysis significantly shortened both the TTR and OS. The median TTR and OS of dogs in relation to the presence of bone lysis was as follows: TTR was 104 days in dogs with bone lysis and 164.5 days in dogs without bone lysis while OS was 139 days in dogs with bone lysis and 259 days in dogs without bone lysis (Figure 3a). However, no significant differences for TTR and OS were shown between melanomas with marrow cavity and without (*p* = 0.98).

In the assessment of response to radiation therapy, twenty-seven (39.7%) dogs achieved a complete response (CR), while a partial response (PR) was observed in 22 dogs (32.4%). Stable disease (SD) was identified in 19 dogs (27.9%). The median TTR and OS for dogs that achieved CR, PR, or SD were as follows: TTR was 195.5, 148, and 90 days in dogs that achieved CR, PR, and SD, respectively, while OS was 308, 200, and 119 days in dogs that achieved CR, PR, and SD, respectively. Moreover, the dogs with bone lysis had a significantly less good response to radiation therapy. The response to radiation therapy in dogs with bone lysis was categorized as follows: SD, 15 dogs; PR, 17 dogs; and CR, 6 dogs. However, there were not any significant differences for response to radiation therapy between melanomas with marrow cavity involvement and without (*p* = 0.46). The corresponding values in dogs without bone lysis were as follows: SD, 3 dogs; PR, 5 dogs; and CR, 21 dogs (*p* = 0.000029). In addition, the dogs with cytoreduction had a significantly better response to radiation therapy. The response to radiation therapy in dogs without cytoreduction was as follows: SD, 18 dogs; PR, 20 dogs; and CR, 8 dogs. On the other hand, the corresponding values in dogs that underwent cytoreduction were as follows: SD, 1 dog; PR, 2 dogs; and CR, 19 dogs (*p* = 0.0000029).

Because clinical staging, tumor size, presence of bone lysis, combination with cytoreduction, and response to radiation therapy significantly affected TTR, multivariate analysis was performed to assess the independent potential prognostic factors affecting TTR and OS. The results showed that clinical stage and SD after radiation therapy were significant prognostic factors (Table 2).

### 3.4. HIF-1α Expression in Melanoma Tissues

Finally, we used IHC to compare the expression of HIF-1α in the intraosseous and extraosseous regions of five melanoma tissues collected using curative intent surgery. The characteristics of the samples used for IHC analysis are listed in Table 3. Although positive staining of the cytoplasm was observed in both intraosseous and extraosseous regions (Figure 4b–d), positive staining of the nucleus was observed more frequently in the intraosseous region than in the extraosseous region (Figure 4e). The positive rates (mean ± SD) in five tissues were intraosseous region, 15.04 ± 1.73; and extraosseous region, 8.36 ± 1.32, respectively. These results indicate that the intraosseous region was under relative hypoxia in comparison with the extraosseous region.

## 4. Discussion

Inoperative canine oral melanoma is often treated with radiation therapy and shows a variable response rate [1]. In the clinical setting of the current study, the positive response rate (CR + PR) to radiation therapy was 72.1%. Radiation therapy generally requires repeated anesthesia and is expensive. Therefore, predicting the response to radiation therapy is important when selecting an appropriate therapy.

This study revealed that advanced clinical stage was significantly associated with a shorter TTR and OS, as suggested by previous studies [3,4]. Median OS in our clinical setting was 144.5 days, which was shorter than that reported in a previous study (233 days) [3]. This may be because the number of dogs with stage I and II disease in this study was fewer than in the previous study (30.9% vs. 41%). Relating to a fewer number of dogs with an early stage, there were no significant differences between TTR and OS in our cohort. Univariate analysis revealed that tumor size, combination with cytoreduction, bone lysis, and response, as well as clinical stage, were significant factors affecting TTR and OS. The tumor location did not affect the efficiency of radiotherapy or OS. On the other hand, clinical stage and response were significant predictive factors based on multivariate analysis.

Whether radiation therapy achieves CR or PR was essential for an improvement in outcomes. The presence of bone lysis and non-combination with cytoreduction significantly worsened the response to radiation therapy, although both the presence of bone lysis and non-combination with cytoreduction were not independent factors affecting the TTR on the basis of multivariate analysis. Altogether, this study suggested that the tumor size had better be reduced prior to radiation therapy to achieve CR or PR.

In our clinical setting, the presence of bone lysis and non-combination with cytoreduction were significantly associated with the progression of the clinical stage, which led to unexpected results in the multivariate analysis. The influence of these factors on TTR should be prospectively evaluated at each clinical stage to exclude the influence of tumor progression.

We evaluated the expression of HIF-1α, a surrogate marker of hypoxia, to estimate hypoxic conditions in several surgically resected oral melanoma tissues. The intranuclear expression of HIF-1α in tumor cells was significantly greater in the intraosseous region than in the extraosseous region, which indicated that the intraosseous region was under hypoxic conditions relative to the extraosseous region. Thus, the presence of local bone lysis might be associated with a negative response to radiation therapy due to hypoxic conditions in tumor tissues. In fact, elevated HIF-1α expression is also known to be associated with a poor prognosis after radiation therapy in human cancers [26,27]. HIF-1α expression is regulated by PI3K/AKT/mTOR signaling, which is suppressed by phosphatase and tensin homolog deletion from chromosome 10 (PTEN) [28]. We recently revealed that PTEN increases the radiosensitivity of canine melanoma cells and is downregulated in recurrent melanoma tissues after radiation therapy in comparison with pretreatment tissues [6]. Altogether, HIF-1α expression may be associated with downregulation of PTEN expression in canine melanoma tissue. Hypoxia and HIF-1α activation are also known to promote tumor progression and metastasis [29,30,31]. Therefore, the relatively rapid regrowth of tumor cells as well as their relative resistance to irradiation due to hypoxia in the osseous tissue in comparison with the periosteum might contribute to the poorer prognosis of dogs with local bone lysis.

The limitations of the current study are as follows: we could not evaluate HIF-1α expression in melanoma tissues treated with radiation therapy. HIF-1α expression should be evaluated in tissues prior to radiation therapy. Furthermore, a larger sample size is needed to reveal the significance of HIF-1α in response to radiation therapy. In addition, there was a potential bias regarding breeds and body weight in our cohort, which might influence the results.

## 5. Conclusions

The clinical stage and response to treatment are important for the efficiency of radiation therapy. In addition, this study suggests that the presence of bone lysis is a significant marker predicting the response to radiation therapy. Furthermore, the tumor size needs to be reduced to improve the response to therapy.

## Figures and Tables

**Figure 1 vetsci-10-00004-f001:**
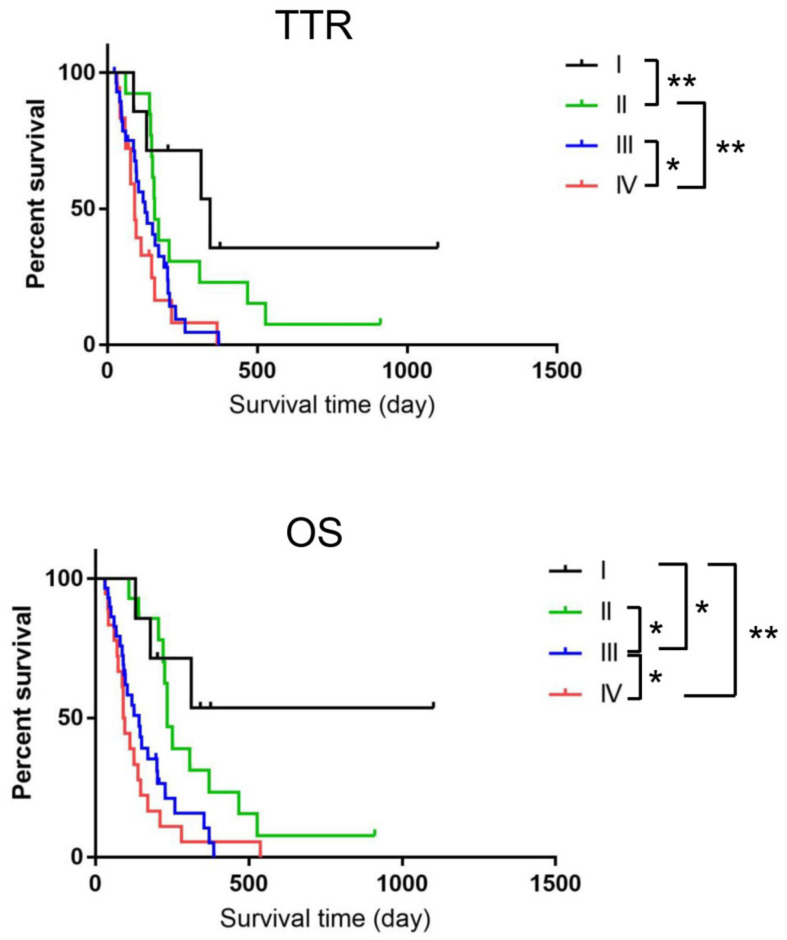
Kaplan–Meier survival curves of the dogs with oral melanoma, evaluated on the basis of World Health Organization tumor stage (I (*n* = 7), II (*n* = 14), III (*n* = 29), and IV (*n* = 18). Significant differences were observed between the groups in brackets. * *p* < 0.05, ** *p* < 0.01.

**Figure 2 vetsci-10-00004-f002:**
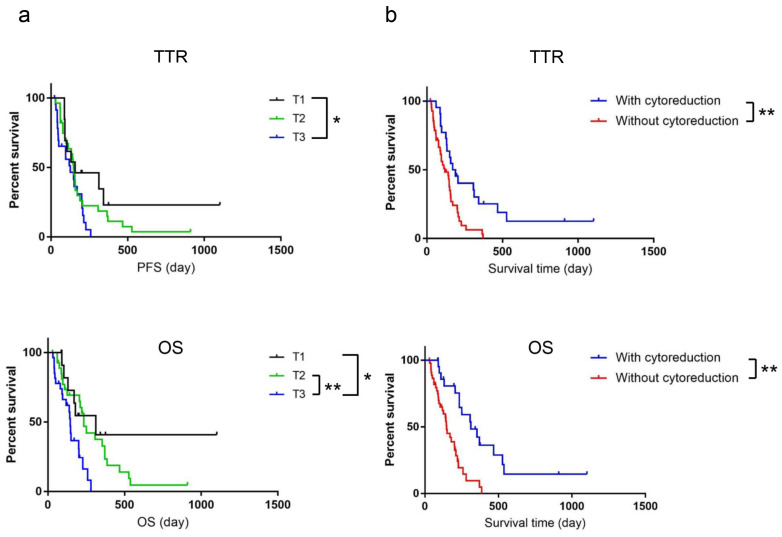
Kaplan–Meier survival curves of dogs with oral melanoma evaluated on the basis of tumor size, ((**a**) T1 (*n* = 13), T2 (*n* = 28), and T3 (*n* = 27)); and cytoreduction, (**b**) with cytoreduction (*n* = 23) and without cytoreduction (*n* = 45). Significant differences were observed between the groups in brackets. * *p* < 0.05, ** *p* < 0.01.

**Figure 3 vetsci-10-00004-f003:**
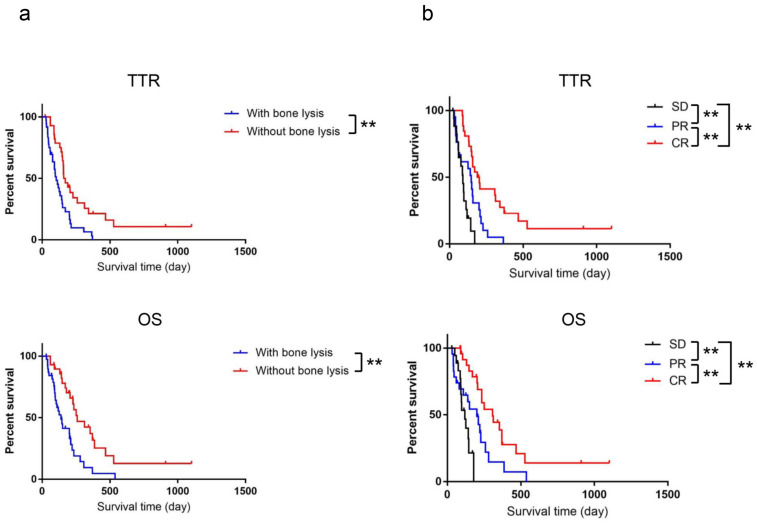
Kaplan–Meier survival curves of the dogs with oral melanoma, evaluated on the basis of the presence of bone lysis, (**a**) with bone lysis (*n* = 39) and without bone lysis (*n* = 29)); and the response to radiation therapy (**b**) SD (*n* = 19), PR (*n* = 23), and CR (*n* = 26)). Significant differences were observed between the groups in brackets. ** *p* < 0.01.

**Figure 4 vetsci-10-00004-f004:**
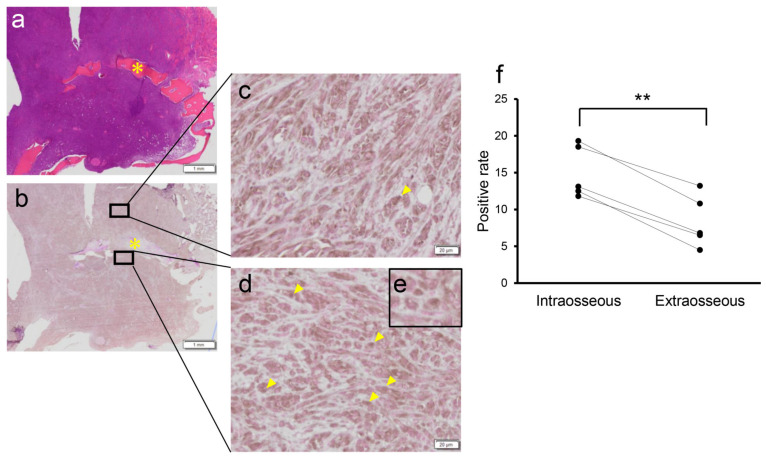
HIF-1α expression analysis by immunohistochemistry in canine melanoma tissues collected by curative intent surgery. (**a**,**b**) Representative image of HE staining (**a**) and immunostaining (**b**) of sample #4; (**c**) A magnified image of the extraosseous region; (**d**) A magnified image of the intraosseous region. Arrowheads indicated positive staining of the nucleus; (**e**) The higher magnification image of HIF-1α positive cell; (**f**) Comparison of HIF-1α expression between the intraosseous and extraosseous regions in five melanoma tissues indicated in Table 3. Significant difference was observed between the groups in brackets. ** *p* < 0.01. Arrow heads indicated HIF-1α positive cells. Bar; 20 µm.

**Table 1 vetsci-10-00004-t001:** Demographic characteristics, anatomic location of oral melanomas, and existence of bone lysis in the oral melanomas of the 68 dogs.

Variable	Stage				*p*-Value
	I (*n* = 7)	II (*n* = 14)	III (*n* = 29)	IV (*n* = 18)	
Median body weight (kg)	7.5	6.3	5.4	4.7	N. S.
Age (years)					N. S.
Median	14	13.5	13	13	
Mean ± SD	13.4 ± 2.2	12.2 ± 3.4	12.7 ± 2.0	13.1 ± 1.6	
Sex					N. S.
Male	4	3	18	4	
Female	3	11	11	14	
Tumor location					N. S.
Maxilla	2	2	13	6	
Mandible	2	9	11	8	
Rostral	1	0	2	2	
Bucca/Lip/Palate	2	3	3	2	
Bone lysis (%)	0 (0)	5 (55.6)	20 (69.9)	14 (77.8)	0.00052
Cytoreduction (%)	5 (71.4)	10 (71.4)	6 (20.7)	1 (5.6)	0.00003

N. S.; not significant.

**Table 2 vetsci-10-00004-t002:** Multivariate Cox proportional hazards regression analysis of predictive factors.

	TTR		OS	
Variable	HR (95% CI)	*p*-Value	HR (95% CI)	*p*-Value
Clinical stage	1.506 (1.038–2.186)	0.0312 *	1.583 (1.026–2.444)	0.038 *
Tumor size	0.972 (0.602–1.570)	0.909	1.158 (0.667–2.012)	0.603
Cytoreduction	0.969 (0.406–2.312)	0.943	1.004 (0.374–2.693)	0.994
Bone lysis	1.057 (0.497–2.244)	0.886	1.164 (0.531–2.552)	0.704
Response	1.869 (1.166–2.997)	0.00937 *	1.825 (1.060–3.140)	0.0299 *

* indicated that the difference is statistically significant.

**Table 3 vetsci-10-00004-t003:** The characteristics of the samples used for IHC.

Sample	Breed	Sex	Age (years)	Body Weight (kg)	Location	Clinical Stage	Bone Lysis
1	Shiba	M	15	9.1	Mandible	II	Yes
2	Toy poodle	SF	12	5	Maxilla	I	Yes
3	Miniature dachshund	SF	12	4.2	Maxilla	III	Yes
4	Toy poodle	SF	15	4	Mandible	III	Yes
5	Mniature schnauzer	SF	8	5.9	Maxilla	III	Yes

M: male, SF: Spayed female.

## Data Availability

The data presented in this study are available on request from the corresponding author.

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
