# Peer review of "Prognostic Factors for the Efficiency of Radiation Therapy in Dogs with Oral Melanoma: A Pilot Study of Hypoxia in Intraosseous Lesions"

_vetsci, 2022, doi:10.3390/vetsci10010004_

Round 1

Reviewer 1 Report

Prognostic factors for the efficiency of radiation therapy in dogs with oral melanoma: A pilot study of hypoxia in intraosseous lesions

Thank you for the opportunity to review this well written and well-presented paper. I have made some suggestions below which would improve the relevance and interest of this study to the reader

Abstract Line 23 Bone lysis and lack of surgery-were associated with less good responses to RT? Rather than “complicated RT”

Abstract Lines 24-27

Immunohistochemical analysis for hypoxia-inducible factor-1α indicated that tumor cells invading the bone are under hypoxic conditions, which is associated with a poorer efficiency of RT in dogs with bone lysis. In conclusion, clinical stage and combination with debulking surgery were needed to improve the efficiency of RT. In addition, canine oral melanoma with bone lysis should be carefully selected for RT because of its resistance to RT”.

In my opinion this is overstated and should be re-worded to avoid misleading the reader. Specifically “tumour cells invading the bone are under hypoxic conditions, which is associated with a poorer efficiency of RT in dogs with bone lysis” sounds as though this has been assessed in vivo when it has not and I suggest is reworded e.g. which may explain the poorer efficiency of RT in dogs with bone lysis.

The last sentence should be removed as it does not add anything to the research and does not consider the potential clinical benefit of RT to the patient (e.g. analgesia)

Methods and Materials

Lines 110-115

Planning target volume margins were contoured to include regions at risk of microscopic disease extension and ranged from 0.5 to 1.0 cm around the gross tumor volume limits”.

This is an unusual way to describe contouring, can the authors please define how the clinical target volume (CTV) and planning target volume (PTV) were decided for macroscopic tumours.

Why was the PTV margin from 0.3-0.5cm as this is usually standardised by the positioning equipment? Was it tumour location/positioning/patient size dependent?

Section 2.3

How was bone lysis defined by what CT characteristics? Did a diplomate in diagnostic imaging read the CT scans?

As one of the objectives of the research is to describe the effect of bone involvement the paper requires much more detail on how this was assessed/measured. It would be interesting to know how many OMM invaded the marrow cavity v just the periosteum

Section 2.4

Were any locoregional lymph nodes irradiated?

Section 3.2

Did any dogs with bone lysis have surgery? If so did this include any removal of bone

Line 180-182

I do not understand what the statistics here refer to? Did the dogs in higher clinical stage have more bone lysis? And patients that had surgery were in lower clinical stage? Could this be reworded please to make the comparison more clear.

What were the differences in stage with bone lysis or not and if surgery or not? It should be clear in the text or minimally refer to a figure.

Section 3.3

Line 230: Moreover, the presence of bone lysis 230 significantly affected the efficiency of radiation therapy. I do not think we can say that the bone lysis directly affected radiation efficiency, rather that dogs with bone lysis had a significantly less good response to radiation therapy. Same for line 234: about cytoreduction, dogs with cytoreduction had better responses to RT

Author Response

Answers to reviewer #1

Thank you for your grateful advisory comments. We did our best to respond to your comments. Thank you for re-review our revised manuscript.

Abstract Line 23 Bone lysis and lack of surgery-were associated with less good responses to RT? Rather than “complicated RT”

>We agreed with your suggestion. According to your recommendation, we revised.

Abstract Lines 24-27

Immunohistochemical analysis for hypoxia-inducible factor-1α indicated that tumor cells invading the bone are under hypoxic conditions, which is associated with a poorer efficiency of RT in dogs with bone lysis. In conclusion, clinical stage and combination with debulking surgery were needed to improve the efficiency of RT. In addition, canine oral melanoma with bone lysis should be carefully selected for RT because of its resistance to RT”.

In my opinion this is overstated and should be re-worded to avoid misleading the reader. Specifically “tumour cells invading the bone are under hypoxic conditions, which is associated with a poorer efficiency of RT in dogs with bone lysis” sounds as though this has been assessed in vivo when it has not and I suggest is reworded e.g. which may explain the poorer efficiency of RT in dogs with bone lysis.

>According to your suggestion, we revised the statement.

The last sentence should be removed as it does not add anything to the research and does not consider the potential clinical benefit of RT to the patient (e.g. analgesia)

>We deleted the last sentence.

Methods and Materials

Lines 110-115

Planning target volume margins were contoured to include regions at risk of microscopic disease extension and ranged from 0.5 to 1.0 cm around the gross tumor volume limits”.

This is an unusual way to describe contouring, can the authors please define how the clinical target volume (CTV) and planning target volume (PTV) were decided for macroscopic tumours.

Why was the PTV margin from 0.3-0.5cm as this is usually standardised by the positioning equipment? Was it tumour location/positioning/patient size dependent?

>As you pointed out, there were some mistypes in the statements. We defined the PTV margin as 0.2 cm in all dogs. On the other hand, the CTV margin was various by tumor location. We revised the statements, based on our actual clinical records.

Section 2.3

How was bone lysis defined by what CT characteristics? Did a diplomate in diagnostic imaging read the CT scans?

>CT images were read by the experienced veterinarian specializing in diagnostic imaging, who was a co-author. The obvious bone destruction alone was defined as bone lysis.

As one of the objectives of the research is to describe the effect of bone involvement the paper requires much more detail on how this was assessed/measured. It would be interesting to know how many OMM invaded the marrow cavity v just the periosteum

>We defined bone involvement as obvious bone destruction. The representative images were added as supplemental figure 1. Of the dogs with bone lysis, our cohort had 28 dogs with invasion into the marrow cavity and 11 with just periosteum. As a result of TTR and OS survival analysis, there were not any significant differences between the dogs with marrow cavity involvement and just periosteum (p = 0.9836). Also, there were no significant differences between the degree of bone lysis and responses to radiation (p = 0.463). These statements were added into section 3.3.

Section 2.4

Were any locoregional lymph nodes irradiated?

>The locoregional lymph nodes with obvious or suspected metastasis were irradiated.

Section 3.2

Did any dogs with bone lysis have surgery? If so did this include any removal of bone

>The melanoma of the dogs, whose tissues were used for IHC, was removed with bone by curative intent surgery. On the other hand, debulking surgery meant removal without bone.

Line 180-182

I do not understand what the statistics here refer to? Did the dogs in higher clinical stage have more bone lysis? And patients that had surgery were in lower clinical stage? Could this be reworded please to make the comparison more clear.

What were the differences in stage with bone lysis or not and if surgery or not? It should be clear in the text or minimally refer to a figure.

>According to your suggestion, we revised the statements.

Section 3.3

Line 230: Moreover, the presence of bone lysis 230 significantly affected the efficiency of radiation therapy. I do not think we can say that the bone lysis directly affected radiation efficiency, rather that dogs with bone lysis had a significantly less good response to radiation therapy. Same for line 234: about cytoreduction, dogs with cytoreduction had better responses to RT

>According to your recommendation, we revised the statements. 

Reviewer 2 Report

The main limitations of the study are mentioned by the authors. In order to contribute to the clarity of the presented results, it is suggested the insertion of photomicrographs with immunostaining for hif-1 at higher magnification.

Author Response

Answer to reviewer #2

Thank you for your grateful advisory comment. We did our best to respond to your comment. Thank you for re-review our revised manuscript.

The main limitations of the study are mentioned by the authors. In order to contribute to the clarity of the presented results, it is suggested the insertion of photomicrographs with immunostaining for hif-1 at higher magnification.

>According to your suggestion, we added a higher magnification image as Fig. 4e.